# Enhanced Nutritional and Functional Recovery in Femur Fracture Patients Post-Surgery: Preliminary Evidence of Muscle-Targeted Nutritional Support in Real-World Practice

**DOI:** 10.3390/geriatrics9060153

**Published:** 2024-11-27

**Authors:** Francisco José Soria Perdomo, Sara Fernández Villaseca, Cristina Zaragoza Brehcist, Elena García Gómez

**Affiliations:** Geriatrics Department, 12 de Octubre University Hospital, 28041 Madrid, Spain; s.fernandez.villaseca@gmail.com (S.F.V.); cristina.zaragoza@salud.madrid.org (C.Z.B.); elegargo@hotmail.com (E.G.G.)

**Keywords:** muscle targeted oral nutritional supplementation, femur fracture, nutritional status, functional capacity

## Abstract

**Background/Objectives**: To describe the effects of muscle-targeted oral nutritional supplementation (MT-ONS) on nutrition, functional capacity, and other health outcomes in patients after femur fracture surgery. **Methods**: A prospective, open-label, single-centre study was conducted. Patients aged 80+ post-femur fracture were recruited. They were assessed at baseline and after 90 days with MT-ONS, 100% whey protein enriched with leucine and vitamin D. Demographics, clinical and nutritional status (MNA^®^-SF), functional capacity [Barthel Index (BI), Lawton and Brody (LB) scale], muscle strength (dynamometry), cognition [Global Deterioration Scale (GDS)], tolerability, and satisfaction data were collected. Descriptive statistics were performed. Ethical approval was obtained. **Results**: Thirty-one patients (74% women, mean age 87 ± 3.99 years) were enrolled. At baseline, 32% were malnourished and 65% were at risk. After ≥90 days of MT-ONS, malnutrition decreased to 13% and well-nourishment increased to 32%. Ninety percent gained weight, with significant muscle strength improvements (+2 kg, *p* < 0.001). Eighty-one percent achieved a BI score ≥ 60 points [mean 84.8 (±17.82)]. BI score improvements correlated with higher baseline muscle strength (rho = 0.413, *p* = 0.021) and better nutritional status (rho = 0.464, *p* = 0.009). The mean LB score was 4.84 (±2.26). Improvements correlated with the pre-fracture BI score (rho = 0.475, *p* = 0.007). Positive correlations were noted between nutritional status, muscle strength, and functional outcomes. Cognition remained stable (GDS = 1 in 67.7% patients). Tolerability and satisfaction with MT-ONS were high at 90%. **Conclusions**: MT-ONS, 100% whey protein enriched with leucine and vitamin D, for ≥90 days enhances nutritional status and functional recovery in patients after femur fracture surgery.

## 1. Introduction

Femur fracture happens in approximately 1 individual per 1000 inhabitants in the majority of European countries [1]. In Spain, about 35,000 femur fractures occur in older people each year, with an incidence of 500–600 cases per 100,000 older patients/year, also presenting much more commonly in women (around 750 cases/100,000 elderly women/year) than in men (around 325 cases/100,000 elderly people/year) [2].

A femur fracture causes approximately 5% to 11% of in-hospital mortality, which varies depending on the average length of stay [3]. Around 15% of patients typically die within three months, with 26–33% succumbing within a year [4]. Further, only 40–60% of patients who survive six months recover the same ability to walk that they previously had; only 40–70% recover their level of independence for basic activities of daily living, and 40% for instrumental activities [5].

Frailty increases the risk of falls and bone fractures [1] and is a predictor of greater pain and functional deterioration after femur fracture [6]. Several studies confirm a strong relationship between frailty, malnutrition, falls, and fractures [1]. Malnutrition is suggested to be one of the main risk factors associated with physical frailty [7]. The European Society for Clinical Nutrition and Metabolism (ESPEN) classifies physical frailty as a nutrition-related condition [8]. A 2021 systemic review found that the prevalence of malnutrition among patients hospitalized for femur fracture to ranges from 4.0% to 39.4% [9]. A meta-analysis showed that malnutrition in patients with femur fracture increased the risk of mortality and morbidity [10]. Providing oral nutritional supplements (ONS) is a common intervention to help meet nutritional needs postoperatively among this population [11]. The ESPEN guidelines provide evidence-based recommendations for clinical nutrition and hydration specifically aimed at preventing and treating malnutrition and dehydration in older patients. The ESPEN 2022 guidelines recommend that older patients with femur fracture should be offered ONS postoperatively to improve dietary intake and reduce the risk of complications (R43, Grade A, strong consensus 100%) [12].

Specialized protein or amino acid supplements can stimulate muscle protein synthesis and improve protein nutritional status, potentially attenuating muscle mass loss, enhancing function, and improving survival in older adults [13]. Leucine, an essential branched-chain amino acid, regulates muscle function [8]. Vitamin D is crucial for calcium absorption, promoting bone health, and reduced femur fracture risk [14]. Leucine-combined supplementation with vitamin D has demonstrated significant benefits for muscle strength and performance, including handgrip strength and gait speed in older adults [15]. In this sense, research evidence shows that muscle-targeted oral nutritional supplementation (MT-ONS), alone or with exercise, is an effective first-line therapy for older patients with sarcopenia, improving clinical outcomes and reducing healthcare costs, especially for those in rehabilitation centres [16].

Postoperative oral nutritional supplementation plays a pivotal role in enhancing femur fracture outcomes by restoring nutritional status to near-optimal levels, preserving functional muscle mass, and reducing complications. However, despite its potential benefits, evidence supporting its effectiveness in routine clinical care is limited, hindering its widespread adoption [17,18]. This study aimed to assess the potential effects of MT-ONS on nutritional and functional changes in patients aged ≥80 years undergoing surgery for femur fracture, with supplementation provided for a minimum of 90 days postoperatively. Additionally, the study sought to evaluate the tolerability and satisfaction levels associated with the ONS.

## 2. Materials and Methods

### 2.1. Design

This study employed a prospective, open-label, descriptive, single-centre, single-arm design involving a cohort of elderly patients who were malnourished or at risk of malnutrition and had experienced femur fractures requiring surgery. These patients received MT-ONS to optimize their caloric and protein intake quality, aiming to improve their nutrition, muscle health, and physical function. Patients were admitted to the Traumatology department at the 12 de Octubre University Hospital (Madrid) and referred to the Geriatric team for a comprehensive assessment.

The primary objective was to assess changes in nutritional status after a minimum of 90 days with MT-ONS from baseline. Additionally, changes in functional capacity, muscle strength, weight, and cognition were evaluated, along with tolerability and satisfaction with the MT-ONS.

### 2.2. Study Population

Patients accomplishing the following inclusion criteria were eligible for entry to the study: age ≥ 80 years; admitted to the Traumatology Service after femur fracture and surgery; referral to the geriatrics team for assessment; and ability to provide written consent for study participation. The exclusion criteria are detailed in Appendix A. Recruitment and assessment.

Participants were consecutively recruited following usual clinical practice, and were assessed at baseline (V1) and at least 90 days after initiation of the MT-ONS (V2) (Figure 1).

Baseline demographic data (age, gender, marital status, place of residence) and clinical data (type of fracture, laboratory tests, nutritional status, cognition, muscle strength, weight, and physical functionality) were collected.

The nutritional status of patients was assessed using the Mini Nutritional Assessment- Short Form (MNA^®^-SF). The MNA^®^-SF test comprises simple measurements: anthropometric measurements (body mass index, weight loss); global assessment (mobility); and dietary questionnaire and subjective assessment (food intake, neuropsychological problems, acute disease). MNA-SF scores of <8, 8–11, and >11 indicate malnourished, risk of malnutrition, and well-nourished, respectively [19,20]. Additionally, the Controlling Nutritional Status (CONUT^®^) score was calculated. The CONUT^®^ score considers serum albumin, total cholesterol level, and total lymphocyte count to assess protein reserves, caloric depletion, and immune performance, respectively, usually compromising undernutrition in hospitalized patients. Based on a screening total CONUT^®^ score, patients are grouped into low risk of malnutrition (0–4 points), moderate risk of malnutrition (5–8 points), and high risk of malnutrition (9–12 points). [21]. Patients’ ability to perform basic activities of daily living was evaluated using the Barthel Index [22,23], while skills of independent living in the community were assessed with the Lawton and Brody scale [24]. Muscle strength was measured using a Jamar hydraulic handheld dynamometer [25], and cognition was documented with the Global Deterioration Scale [26].

Tolerability and satisfaction with the ONS were determined at V2. Participants rated their levels of tolerability and satisfaction on a Likert-type scale ranging from 1 to 10, with 1 indicating very low tolerance/satisfaction and 10 indicating very high tolerance/satisfaction.

### 2.3. Oral Nutritional Supplementation

The ONS provided was hypercaloric (151 kcal/100 mL) and hyperproteic (10.4 g/100 mL) 100% serum lactoprotein enriched with leucine and vitamin D (Appendix A). Patients were instructed to take one to three bottles (200 mL/bottle) per day, depending on individual needs.

### 2.4. Statistical Analysis

Descriptive statistics were applied. Percentages, mean, standard deviation, median, interquartile range, and minimum and maximum range were calculated. The comparison of two dependent variables was carried out using the Student’s *t*-test in the case of a normal distribution. Otherwise, the non-parametric Wilcoxon–Mann–Whitney test was employed.

Spearman’s correlation was used to assess the correlation between nutritional status and baseline characteristics (age, Barthel Index, Lawton and Brody scale, and muscle strength). The correlation between the nutritional status (MNA^®^-SF) and mean days of ONS was also assessed. Additionally, correlations between the change in nutritional status and the change in other outcomes (Barthel Index, Lawton and Brody scale, muscular strength, levels of albumin, cholesterol, and lymphocyte count) were also calculated.

The regression analysis was conducted by employing the Recursive Partitioning and Regression Trees (RPART) algorithm [27]. It is a machine learning technique that builds decision models from data by iteratively splitting datasets based on predictor values to maximize prediction accuracy. It was conducted to explore the baseline characteristics influencing post-ONS nutritional status based on MNA^®^-SF score and post-ONS independence to perform basic activities of daily living based on Barthel Index score. The potential influencing factors considered were baseline Barthel Index score, Lawton and Brody scale score, weight, muscle strength, age, MNA^®^-SF score, gender, type of fracture, and mean number of days on MT-ONS. The objective was to identify groups of patients more likely to experience changes in nutritional status and independence after at least 90 days on MT-ONS based on their baseline characteristics [28]. Data were analyzed with JASP 0.18.3.

### 2.5. Ethical Considerations

Patients received an information sheet outlining the study’s name, objectives, sponsors, data anonymization and processing, and voluntary participation. All patients signed a written consent form. The study was approved by the Clinical Research Ethical Committee of the University Hospital 12 de Octubre in Madrid, Spain (CEIm Number 21/543).

## 3. Results

### 3.1. Baseline Demographic and Clinical Characteristics at Visit 1

A total of 31 patients were recruited between 10th February 2022 and 24th August 2023. The mean age was 87 years (±Standard deviation, SD) (±3.99), and 74% were women; 77% were widows; 68% had a trochanteric fracture and no cognitive decline (67.7%). Before being admitted to hospital, 78% lived in their own homes, either with (36%) or without (42%) a contracted caregiver and returned home at hospital discharge (Table 1).

At V1, 32% of participants were malnourished, 65% were at risk of malnutrition, and 3% were well nourished according to the MNA^®^-SF score (Table 2). The mean MNA^®^-SF score for the entire population was 8.6 (±2.01). According to CONUT^®^, 3% were at high risk of malnutrition, 42% at moderate risk of malnutrition, and 55% at low risk of malnutrition (Table 2). The median weight was 60.4 kg (±9.96). Albumin levels, total cholesterol levels, and lymphocyte count were at the lower limits of the normal ranges. The mean Barthel Index score was 90.97 (±13.57), and 96.8% had a Barthel Index ≥60 on the day before the fracture. The mean Lawton and Brody score was 5.13 (±2.18). On the day before the fracture, patients were either independent (29%) or partially (35.5%) or very dependent (22.6%) in performing instrumental activities of daily living (Table 2).

### 3.2. Clinical Characteristics at Visit 2

The average time between V1 and V2 while on MT-ONS was 122 days (±92.08). All participants mentioned having followed the nutritional recommendations provided by the healthcare team (Appendix A). Further, 15 out of 31 patients improved their nutritional status according to the MNA^®^-SF score. The presence of malnutrition decreased to 13% while the proportion of well-nourished patients increased to 32%; 55% remained at risk of malnutrition (Figure 2). The mean score on the MNA^®^-SF increased by 1.8 points, from 8.6 (±2.01) to 10.4 (±2.38). Additionally, 97% of the patients were classified as being at low risk of malnutrition according to CONUT^®^ at V2.

We found that 90% (28/31) experienced weight gain or maintained their weight, with a mean weight gain of 2 kg (±0.68). The mean muscle strength gain measured with dynamometry was 2kg (±0.043). Albumin levels, total cholesterol levels, and lymphocyte count increased, surpassing the inferior limits of the previous ranges (Figure 3a). Cognition remained unaltered and preserved in 67.7% of participants, as observed at V1 (Table 2).

Additionally, 81% of patients achieved a Barthel Index score of ≥60 after the femur fracture and while receiving MT-ONS. The Barthel Index score was almost restored to the V1 score, averaging 84.8 (±17.82) at V2 (Figure 3b). Although most activities of daily living were compromised after the femur fracture, except for the ability to bath, the item analysis of the Barthel Index revealed that the greatest decline happened in the ability to climb stairs (−1.29 ± 2.61) and the ability to transfer from bed to chair and back (−1.29 ± 2.32). Likewise, independence to perform instrumental activities of daily living was almost restored after an average of 122 days with MT-ONS, with a mean score on the Lawton and Brody scale of 4.84 (±2.26) at V2.

### 3.3. Drivers of Changes in Nutritional Status

A positive association was observed between an improvement in nutritional status, as indicated by the MNA^®^-SF at V2, and the Barthel Index score prior to the fracture (rho = 0.432, *p* = 0.015) (Appendix A). This suggests that patients with better nutritional status at V2 had enhanced functionality in independently performing activities of daily living before the fracture. Likewise, better nutritional status according to the MNA^®^-SF score was positively associated with increases in serum albumin levels (rho = 0.379, *p* = 0.036), total cholesterol levels (rho = 0.452, *p* = 0.011), and the Lawton and Brody score (rho = 0.410, *p* = 0.022) at V2 (Appendix A). This implies that individuals with improved nutritional status at V2 experienced enhancements in their nutritional intake, as well as in their functionality to independently perform instrumental activities of daily living.

### 3.4. Factors Predicting Nutritional Improvement with MT-ONS

The regression analysis employing a decision tree based on the RPART algorithm [26] demonstrated 77.4% precision, indicating that patients with a Barthel Index score >90 and a Lawton and Brody score <5 on the day prior to the fracture were more likely to change to a group with better nutritional status after at least 90 days on MT-ONS (R2 = 0.37; accuracy = 77.4%). Approximately 37% of the variability in the outcome (changes in nutritional status assessed with the MNA^®^-SF) could be explained by the factors considered in the analysis (Figure 4a). This suggests that patients with better functional independence in basic and instrumental activities of daily living on the day prior to the fracture improved their nutritional status while on MT-ONS.

### 3.5. Drivers of Changes in Independence to Perform Activities of Daily Living

Changes in the Barthel Index score at V2 were positively correlated with baseline higher muscle strength (rho = 0.413, *p* = 0.021) and better nutritional status according to MNA^®^-SF score (rho = 0.464, *p* = 0.009) (Appendix A). The change in muscle strength in V2 was associated with a Barthel Index total score ≥60 in V1 (rho = 0.375, *p* = 0.038), implying that individuals with better baseline functional status experienced improvements in muscle strength with the MT-ONS. Improvements in the Lawton and Brody scale were positively associated with the total score on the Barthel Index on the day before the fracture (rho = 0.475, *p* = 0.007) (Appendix A). This suggests that patients who demonstrated greater independence in basic daily activities at V2 also exhibited higher muscle strength and better nutritional status at V1. Similarly, those with better functionality in instrumental activities of daily living were more independent in basic daily living tasks before the fracture.

### 3.6. Factors Predicting Independent Functionality Improvement with MT-ONS

The regression analysis, utilizing a decision tree based on the RPART algorithm [26], demonstrated a 71% precision. This indicates that patients with muscle strength ≥11 kg on dynamometry and a Lawton and Brody score <7 the day prior to the fracture were more likely to experience improvement in their Barthel Index score after at least 90 days on MT-ONS (R2 = 0.34; accuracy = 71%) (Figure 4b) and to recover their Barthel Index score prior to the fracture. Approximately 34% of the variability in the improvement of Barthel Index scores can be explained by these factors. This suggests that higher muscle strength (higher scores on dynamometry) and a lower capacity for performing instrumental activities of daily living before the fracture (lower scores on Lawton and Brody scale) predict positive changes in independence to perform basic self-care activities of daily living (higher Barthel Index scores).

### 3.7. Tolerability and Satisfaction with the MT-ONS

Further, 91% of participants reported good (Likert scale score: 7 to 8) or very good (Likert scale score: 9 to 10) tolerability. No nausea, vomiting, diarrhea, constipation, or reflux were reported. Additionally, 90% reported high (Likert scale score 7 to 8) or very high satisfaction (Likert scale score: 9 to 10) with the MT-ONS (Appendix A).

## 4. Discussion

The results of this prospective clinical study show changes in the nutritional status of elderly patients aged over 80 years who experienced femur fractures requiring surgery and who were either malnourished or at risk of malnutrition. These patients received a hypercaloric and hyperproteic MT-ONS, 100% whey protein enriched with leucine and vitamin D, for a minimum of 90 days to optimize their caloric and protein intake. They were referred to the geriatric team for a comprehensive assessment at the hospital’s traumatology department as part of routine medical practice.

The results show that the frequency of malnutrition risk in this study is slightly higher compared to previously reported studies [29,30], and that a noticeable shift towards a well-nourished status (from 3% to 32%) and an increase in the percentage of patients at low risk of malnutrition (from 55% to 97%) is observed after a minimum of 90 days of MT-ONS, as assessed by the MNA^®^-SF and CONUT^®^ screening tools. This suggests that MT-ONS may effectively contribute to the improvement of these patients’ nutritional status. Similar results were also observed in another clinical study involving malnourished femur fracture patients receiving MT-ONS at an Orthogeriatric Unit that also showed improvements with early and continued nutritional support [31]. However, 55% of patients remain at risk of malnutrition according to the MNA^®^-SF, highlighting the need for attention to prevent further deterioration of their nutritional status, or to continue nutritional recovery. These findings align with the conclusions of recent systematic reviews, which suggest that MT-ONS administration in femur fracture patients can recover nutritional status to near-optimum levels, may reduce postoperative complications, and improve functional status [11,17].

Body weight, as well as known markers of nutritional status such as albumin and total cholesterol, increased in this study population. Additionally, muscle strength showed improvement, further reinforcing the positive effects of MT-ONS on patients’ nutritional status, consistent with previously reported evidence [30,31,32,33]. In previous studies, higher grip strength was linked to an improved ability to perform daily activities [34], while lower handgrip strength upon admission was associated with increased dependency in basic [35] and instrumental activities of daily living [36]. In this study, a moderate correlation was found between greater initial muscle strength and the recovery of pre-fracture Barthel Index scores, indicating improved functional outcomes. Therefore, in the context of clinical nutrition for the elderly, these findings suggest that factors such as higher muscle strength (≥11), reflected in dynamometry scores, are associated with greater functional recovery. This implies that interventions aimed at enhancing muscle strength and promoting muscle health may contribute to improved functional recovery in elderly patients [35,37]. Muscle strength in the elderly is also associated with increased independency in daily living and better quality of life [36,38].

In this study, the MT-ONS provided 100% whey protein, a high-quality protein source enriched with essential amino acids crucial for maintaining muscle mass and function in elderly individuals [39,40]. Additionally, the inclusion of leucine, an essential amino acid known for its role in stimulating muscle protein synthesis, enhances the anabolic effect of serum lactoprotein, facilitating muscle preservation and strength maintenance in the elderly [40]. This combination may be particularly beneficial for elderly individuals susceptible to sarcopenia or muscle wasting, as preserving or restoring physical functionality is of the utmost importance [41]. Recent intervention studies across various healthcare settings support muscle-targeted oral nutritional supplementation to improve muscle mass, function, and physical performance in sarcopenia patients. Greater efficacy is seen when combined with tailored resistance exercise and achieving adequate protein-calorie balance. Proactive application is also beneficial, as evidence shows maintenance or improvement in patients at risk of muscle loss [42].

Most patients demonstrated high levels of independence in daily living and instrumental activities before experiencing a femur fracture, as evidenced by their pre-fracture Barthel Index and Lawton and Brody scale scores. These high pre-fracture scores are essential for achieving complete functional recovery after fracture [43]. In this study, 81% of individuals regained their pre-fracture functional status for basic self-care tasks post-ONS, a larger proportion compared to previous studies where lower pre-fracture Barthel Index scores were associated with higher dependence and slower recovery [44,45]. The sustained lower capacity for instrumental activities of daily living, as indicated by Lawton and Brody scores (<7) maintained by more than half of patients after femur fracture and despite MT-ONS, suggests that focusing on maintaining independence in basic activities of daily living while providing assistance for more complex tasks may facilitate overall functional improvement in elderly individuals undergoing clinical nutrition interventions [46].

Furthermore, patients with better functional independence in both basic and instrumental activities of daily living prior to femur fracture were more likely to achieve greater improvements in nutritional status post-ONS. The identified factors, such as muscle strength on dynamometry, Barthel Index, and Lawton and Brody scores before the fracture, are significant predictors of basic functional independence and nutritional status post-ONS. However, the models accurately predict improvements in only about 70% of cases and explain approximately one-third of the variability in outcomes. This implies that other factors, not included in the analysis, may also contribute to the improvement in the nutritional and functional conditions of the patients. Consideration of multiple interconnected factors is essential in clinical nutrition interventions for promoting functional recovery in elderly individuals with femur fractures [47,48,49,50]. The relationship observed between these factors and functional outcomes stresses the potential effectiveness of tailored interventions aimed at preserving muscle health and independence in daily activities to optimize outcomes in this population.

The results of this study also strengthen the importance of early interventions, even in individuals with Barthel Index scores > 90 and as young as 65 years of age, which are crucial to maintain functional independence and to achieve superior recovery rates in terms of nutritional status. Interventions incorporating physical exercise with supplementation, cognitive training, or supplementation alone have yielded favourable effects on frailty indicators. Economic analyses have shown that these interventions provide better value for money compared to standard care, particularly benefiting very frail individuals living in the community. Moreover, they have demonstrated positive impacts on frailty-related outcomes in both hospital and outpatient settings, all without increasing costs or maintaining cost-effectiveness. However, individuals with acceptable functionality scores are often overlooked in frailty prevention initiatives or deemed exempt from interventions. For instance, the frailty prevention programme implemented by the Community of Madrid, in Spain, for individuals aged 70 years and older excludes those with Barthel Index scores > 90 [51].

This study presents several limitations. Its descriptive design, lacking a control group, hinders drawing definitive conclusions and limits the robustness of the findings. Conducted solely in a single hospital, its scope may limit reliability. The lack of multiple testing analyses increases the risk of false positives. The small sample size and challenges in recruiting older adults with femur fractures for nutritional assessment limit the robustness, generalizability, and statistical power of the findings, highlighting the need for confirmation in clinical studies with a more robust design. The study assesses the short-term effects of the MT-ONS benefits, leaving the long-term impact of its benefits not yet fully understood. Future research should focus on long-term follow up to understand sustained effects given the chronic nature of nutritional and functional decline in older adults. However, despite these limitations, the study also possesses strengths. It accurately reflects clinical practises at a tertiary hospital and stresses the beneficial effects of MT-ONS on the nutritional and functional recovery of old patients with femur fracture and surgery. Additionally, it emphasizes the importance of early and sustained intervention for optimal nutritional outcomes and for maintaining high functional independence in the elderly. This study provides preliminary evidence for further confirmatory research.

## 5. Conclusions

This study demonstrated improvements in nutritional status and functional outcomes among elderly patients with femur fracture and surgery who received MT-ONS 100% whey protein enriched with leucine and vitamin D for a minimum of 90 days. MT-ONS contributed to decreasing malnutrition, increasing the proportion of well-nourished individuals, and promoting weight gain. Improvements in muscle strength and functional independence correlated with baseline scores and nutritional status, suggesting the importance of early intervention. However, a high percentage of patients remained at risk of malnutrition, and the slower recovery of independence in performing instrumental activities of daily living in the community emphasizes the need for sustained support. High tolerability and satisfaction with the MT-ONS further endorse its effectiveness.

## Figures and Tables

**Figure 1 geriatrics-09-00153-f001:**
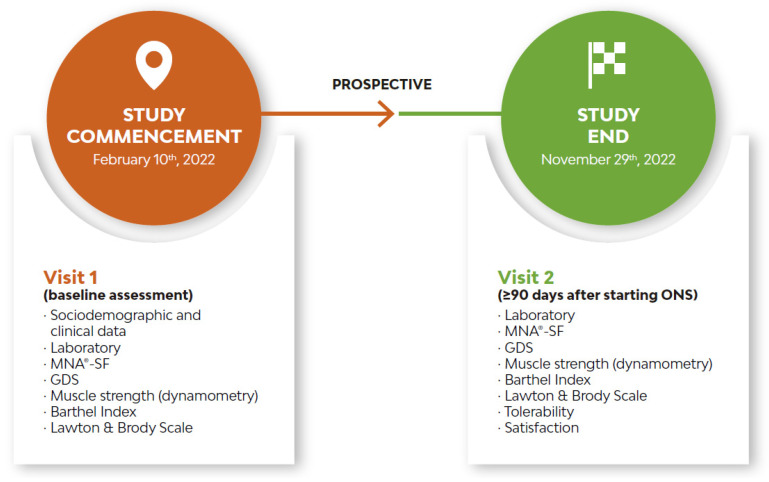
Study design. GDS: Global Deterioration Scale, MNA^®^-SF: Mini Nutritional Assessment Short Form, ONS: oral nutritional supplementation.

**Figure 2 geriatrics-09-00153-f002:**
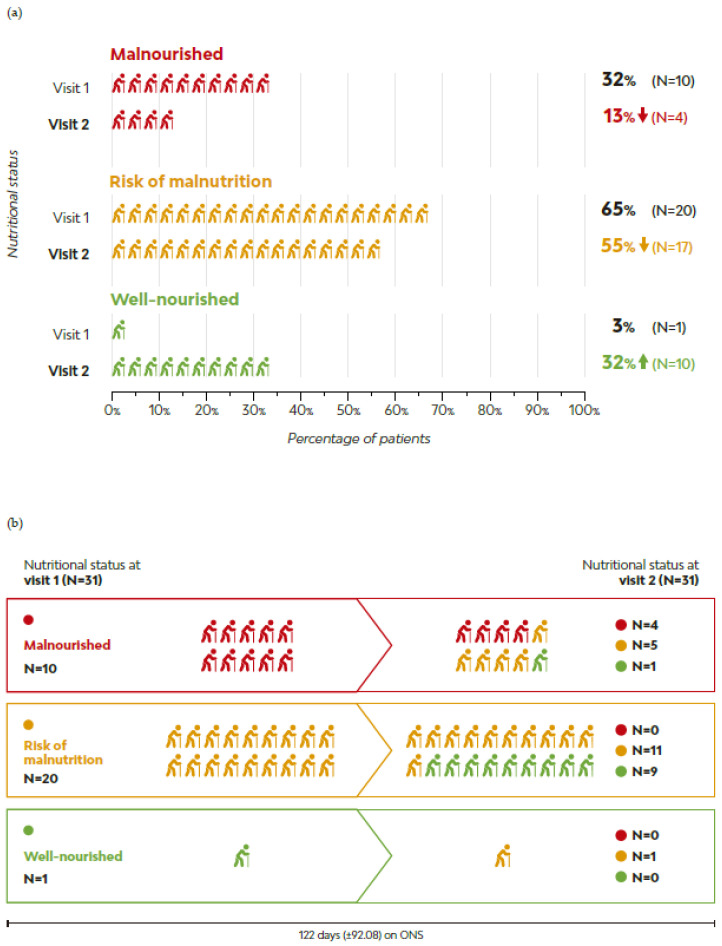
Nutritional status distribution across visits: MNA^®^-SF scores at visit 1 (baseline) and visit 2: (**a**) Nutritional status distribution in percentages. (**b**) Number of patients in each MNA^®^-SF group at visit 1 and visit 2. Shift in the number of patients in each MNA^®^-SF group at visit 1 and visit 2: Top line: 10 patients who were malnourished (red) at visit 1 were assessed at visit 2: 4 patients remained malnourished (red), 5 were at risk of malnutrition (yellow) and 1 well-nourished (green). Middle line: out of a total of 20 patients who were at risk of malnutrition at visit 1 (yellow), 11 remained at risk of malnutrition (yellow) and 9 were well-nourished (green). Bottom line: 1 patient was well-nourished (green) at visit 1 and at risk of malnutrition (yellow) at visit 1.

**Figure 3 geriatrics-09-00153-f003:**
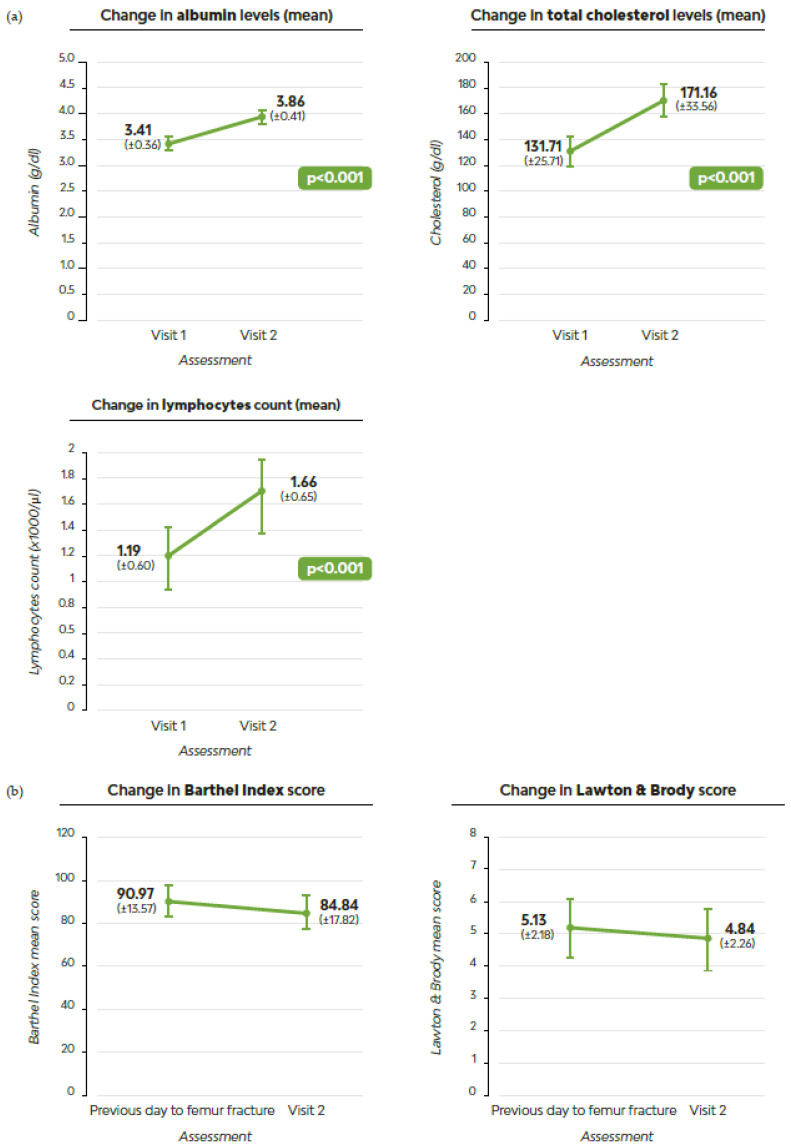
Comparison of laboratory parameters, Barthel Index, and Lawton and Brody scale scores between visit 1 or pre-fracture and visit 2: (**a**) Laboratory parameters; (**b**) Barthel Index and Lawton and Brody scale scores.

**Figure 4 geriatrics-09-00153-f004:**
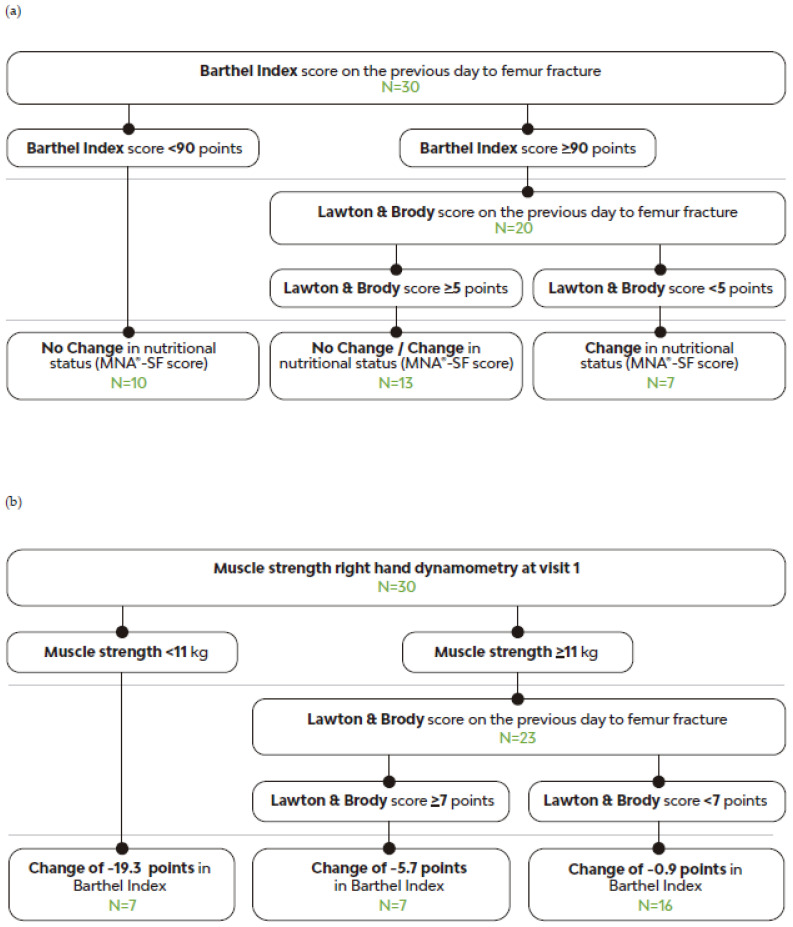
Decision tree algorithms: (**a**) Decision tree algorithm for changes in nutritional status based on MNA^®^-SF scores. (**b**) Decision tree algorithm for the change in Barthel Index. No change: patients remained in the same nutritional status group according to MNA^®^-SF at visit 1 and visit 2. Change: patients changed nutritional status group according to MNA^®^-SF from visit 1 to visit 2 (R2 = 0.37; accuracy = 77.4%); MNA^®^-SF: Mini Nutritional Assessment Short Form. Barthel Index scores decreased after femur fracture but progressively recovered over time and with ONS approaching pre-fracture values. Patients with higher muscle strength at visit 1 and lower Lawton and Brody scores pre-fracture had higher Barthel Index scores, exhibiting a smaller decrease at visit 2 compared to pre-fracture values (R2 = 0.34; accuracy = 71%); ONS: oral nutritional supplementation.

**Table 1 geriatrics-09-00153-t001:** Baseline demographic characteristics of the study population.

Characteristics	
Age, years (±SD)	87 (±3.9)
Gender	
Female, *n* (%)	23 (74%)
Male, *n* (%)	8 (26%)
Marital status	
Single, *n* (%)	0 (0%)
In a relationship, *n* (%)	0 (0%)
Married, *n* (%)	7 (23%)
Divorced, *n* (%)	0 (0%)
Widow, *n* (%)	24 (77%)
Habitual residence location	
Home, alone without a contracted caregiver, *n* (%)	13 (42%)
Home, alone with a contracted caregiver, *n* (%)	12 (36%)
Home, with a family member, *n* (%)	5 (16%)
Home, with a family member and a contracted caregiver, *n* (%)	1 (3%)
Home of a family member, relative or friend, *n* (%)	1 (3%)
Institution	0 (0%)
Residence location at hospital discharge	
Habitual residence	24 (78%)
Habitual institution	1 (3%)
Intermediate care centre	6 (19%)
Nursing home	0 (0%)
Others	0 (0%)

**Table 2 geriatrics-09-00153-t002:** Study population clinical characteristics at visit 1 (baseline) and visit 2.

Characteristics	Visit 1	Visit 2
Type of fracture		
Trochanteric	21 (68%)	-
Sub-trochanteric	1 (3%)	-
Sub-capital	9 (29%)	-
MNA^®^-SF score		
12–14 points, well-nourished, *n* (%)	1 (3%)	10 (32%)
8–11 points, at risk of malnutrition, *n* (%)	20 (65%)	17 (55%)
0–7 points, malnourished, *n* (%)	10 (32%)	4 (13%)
Laboratory parameters		
Serum albumin levels (g/dL) mean (±SD)	3.4 (±0.36)	3.9 (±0.41)
Cholesterol levels (md/dL) mean (±SD)	131.7 (±25.71)	171.2 (±33.56)
Lymphocyte count (×1000/µL) mean (±SD)	1.2 (±0.60)	1.7 (±0.65)
CONUT^®^		
9–12 points, high risk of malnutrition, *n* (%)	1 (3%)	0 (0%)
5–8 points, moderate risk of malnutrition, *n* (%)	13 (42%)	1 (3%)
0–4 points, low risk of malnutrition, *n* (%)	17 (55%)	30 (97%)
Weight, kg (SD)	60.4 (±9.9)	62.4 (±9.2)
Muscle strength right hand, kg (SD)	15.7 (±5.0)	17.1 (±5.1)
Muscle strength left hand, kg (SD)	14.2 (±5.1)	15.1 (±4.2)
Cognitive status		
GDS 1: no cognitive decline, *n* (%)	21 (67.7%)	21 (67.7%)
GDS 2: very mild cognitive decline, *n* (%)	2 (6.5%)	3 (9.7)
GDS 3: mild cognitive decline, *n* (%)	5 (16.1%)	5 (16.1%)
GDS 4: moderate cognitive decline, *n* (%)	3 (9.7%)	2 (6.5%)
GDS 5: moderately severe cognitive decline, *n* (%)	0% (0)	0% (0)
GDS 6: severe cognitive decline, *n* (%)	0% (0)	0% (0)
GDS 7: very severe cognitive decline, *n* (%)	0% (0)	0% (0)
Barthel Index score (previous day to femur fracture)		
100 points (independent in the daily activities), *n* (%)	15 (48.4%)	13 (41.9%)
60–95 points (needs minimal help with daily activities), *n* (%)	15 (48.4%)	14 (45.2%)
45–55 points (partially dependant), *n* (%)	1 (3.2%)	4 (12.9%)
20–40 points (very dependant), *n* (%)	0 (0%)	0 (0%)
<20 points (totally dependent), *n* (%)	0 (0%)	0 (0%)
Lawton and Brody scale scores (previous day to femur fracture)		
0–1 point (totally dependent), *n* (%)	1 (3.2%)	0 (0%)
2–3 points (very dependant), *n* (%)	7 (22.6%)	12 (38.7%)
4–5 points (partially dependant), *n* (%)	11 (35.5%)	8 (25.8%)
6–7 points (low dependence), *n* (%)	3 (9.7%)	3 (9.7%)
8 points (independent), *n* (%)	9 (29.0%)	8 (25.8%)

CONUT^®^: Controlling Nutritional Status, GDS: Global Deterioration Scale, MNA^®^-SF: Mini Nutritional Assessment Short Form, SD: standard deviation.

## Data Availability

Data are contained within the article and Appendix A.

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
