# Peer review of "Enhanced Nutritional and Functional Recovery in Femur Fracture Patients Post-Surgery: Preliminary Evidence of Muscle-Targeted Nutritional Support in Real-World Practice"

_geriatrics, 2024, doi:10.3390/geriatrics9060153_

Round 1

Reviewer 1 Report

Comments and Suggestions for Authors

Thank you for the opportunity to review this manuscript, which presents a prospective, open-label study investigating the effects of muscle-targeted oral nutritional supplementation (MT-ONS) on elderly femur fracture patients. Its prospective design, focusing on elderly patients with femur fractures, addresses an important clinical population where malnutrition and muscle weakness significantly impact recovery.

Below I tried to review areas for improvement and recommendations point-by-point.

Introduction
The introduction is clear and covers the topic sufficiently

Material and Methods

We also use nutritional supplements for elderly patients with hip fractures, but we are mostly confronted with the problem that its really hard to observe (in the daily routine) if the patients did fully drink their supplement. How was that done in this study? 

Statistics

Did the authors conduct any multiple testing, if yes please mention if any correction methods (e.g., Bonferroni correction) were applied. If none were used, this should be noted.

Most of readers wont be familiar with regression analysis using RPART. Therefore i would suggest to explain the model and give more information about the interpretation of 37% and especially in means of clinical terms. Furthermore the low R² values could also suggest that the model does not fully capture the variability in the outcomes. If yes, then the authors the should explore other factors that could account for the remaining variability

The results mention several correlations between baseline muscle strength and changes in the Barthel Index score, nevertheless they suggest moderate relationships at best. I suggest to mention this in the discussion.

Discussion

I suggest to include a more detailed comparison to recent meta-analyses or systematic reviews on the role of nutritional supplementation in elderly fracture patients. The observed improvements in muscle strength (+2 kg) are significant, but the clinical importance of this change is not fully contextualized, therefore this should be shortly dealt with.

Conflict of Interest and Funding

Line 403 "Danone Nutricia SL provided financial support to develop the study and write the manuscript."

I strongly advise the authors to either comment or correct this statement. How far was the financial support and in which way has the writing of the manuscript been supported? I suggest to include a statement detailing the extent of the sponsor's involvement.

Reviewer 2 Report

Comments and Suggestions for Authors

The article titled “Enhanced nutritional and functional recovery in femur fracture patients’ post-surgery: preliminary evidence of muscle-targeted nutritional support in real-world practice” by Fernández Villaseca et al. investigates the effects of muscle-targeted oral nutritional supplementation (MT-ONS) on nutrition, functional capacity, and health outcomes in older patients post-femur fracture surgery. Results indicate improved nutritional status, functional recovery, and muscle strength, with high patient satisfaction. This study emphasizes the potential role of MT-ONS in post-operative recovery, though it is limited by a small sample size and lack of a control group.

Major Issues

  1. The lack of a control group in this open-label study design limits the robustness of the findings. Without comparison to a non-supplemented group, attributing improvements solely to MT-ONS is challenging. If feasible, including a control group would strengthen the study’s findings. If not, the authors should elaborate on why this was not possible and further discuss how this limitation might impact the study's validity.
  2. The study’s small sample size (n=31) limits generalizability and statistical power, making it difficult to apply findings to a larger population confidently.
  3. While descriptive statistics were presented, the study would benefit from a more rigorous statistical analysis, such as multivariate regression or ANCOVA, to account for potential confounders.
  4. Discuss the need for longer follow-up studies to assess the sustainability of MT-ONS benefits and suggest this as a future research direction.
  5. The study assesses only short-term effects (90 days). Given the chronic nature of nutritional and functional decline in older adults, long-term follow-up is essential to understand sustained effects.

Minor issues

1.        Some descriptions, such as the CONUT score and MNA-SF classifications, could be better explained for readability. Adding context on why certain thresholds were chosen would improve clarity.

2.        Some references, particularly guidelines cited, could be further contextualized. Including more recent studies or guidelines might strengthen the background section.

3.        Please correct typographical errors 

Round 2

Reviewer 1 Report

Comments and Suggestions for Authors

Line 474: Thank you for the changes you have made so far. Unfortunately, the current information on sponsorship is still insufficient. I suggest to  provide a list in the supplement section in the form of keywords, indicating the extent to which and for what the sponsorship was received and or used.

Reviewer 2 Report

Comments and Suggestions for Authors

The authors adequately addressed the reviewers' recommendations. I have no further comments.
